# Comparison of the Transition to More Sustainable Stormwater Management in China and the USA

Yitong Zhao *, Mackay Price and Sam Trowsdale

Department of Environmental Science, University of Auckland, Auckland 1010, New Zealand;
wpri344@aucklanduni.ac.nz (M.P.); s.trowsdale@auckland.ac.nz (S.T.)
* Correspondence: yzhb454@aucklanduni.ac.nz; Tel.: +64-273-094-241

**Abstract:** This paper presents a comparative cross-nation study of the transition to more sustainable stormwater management (SSWM) in the United States and China. Multi-level perspective and multiphase models are used to examine the transition dynamics and reflect on how transition theory explains the change within federal and socialist context. Instead of simply differentiating the two countries' transition patterns by using terms such as bottom-up or top-down, we consider the importance of changes at all three levels of the system. The main difference between the transition process in the United States and China is the extent to which niche level innovations are developed, especially in the type of actors and activities investigated. The analysis suggests that the Chinese transition is less radical, while the U.S. pathway exhibits signs of reconfiguration, dealignment and realignment. Developing learning networks across sectors and actors to spread knowledge and experience appears to be the next major challenge for the Chinese Sponge City initiative. Despite the feasibility of transition theory for transition comparison, the author suggests its usage with caution and critical reflection to avoid the risk of embedding the mindset of 'catch-up' and convergence.

**Keywords:** stormwater management; transition; multi-level perspective

## 1. Introduction

Urban stormwater management is traditionally based on the principles of a command-and-control technocracy whereby water is transported away from a city as quickly as possible using built infrastructure to avoid flooding. Building on the sustainability debate that came to the fore in the 1990's, there was a widespread aspiration to shift to more sustainable stormwater solutions. These took off at the turn of the 21st Century with strategies such as Water Sensitive Urban Design (WSUD), Sustainable Urban Drainage Systems (SuDs), and Low Impact Development (LID) and gained global traction as the world urbanized and looked to do so in an environmentally friendly way [1]. In a similar way, the Sponge City Initiative (SCI) was proposed by the Chinese government in 2012, envisioning cities with the capacity to infiltrate, drain and filter stormwater freely, improving resilience to floods, droughts, and contamination, hooking into some of the major problems in the contemporary environmental crisis discourse [2].

The SCI is a national strategy to achieve the goals of new-type urbanization (xinxingchengzhenhua) and the harmonious development of human and nature [3]. SCI calls for more integrated urban water management rooted in the physical and hydrosocial water cycle [4,5] and a planning and design strategy for sustainable urban development that promotes rainwater systems integrated with strategic ecosystem conservation and restoration or remediation [6]. Its implementation requires a comprehensive integration of multiple aspects from policies, designs, to social communication and other subsystems [7] and in many ways contrasts to the techno-centric six-word principles (of the sponge city) that have been frequently adopted by the government, academy and industry (infiltration, detention, storage, purification, usage, and drainage).

Perhaps the most familiar change to stormwater management in the SCI is the encouragement of LID as an alternative approach (albeit largely technical) to hard-engineering stormwater systems, primarily for flood mitigation and pollution control. Borrowing the terminology from the U.S., LID refers to on-site natural and engineered infiltration and storage techniques such as pervious paving, rain-gardens, and greenroofs. However, widespread implementation of new approaches is not easy because established systems tend to privilege previous technologies and practice, known as path dependency.

Addressing such persistent societal problems (often environmental) is a central topic of socio-technical transition research. From a socio-technical perspective, sustainability transitions cannot be achieved through 'green' technology alone but requires a broader systemic societal change, whereas technical evolution can be used as an entrance point for studying societal changes [8,9]. Sustainable stormwater management (SSWM) transition, in-line with other subjects, refers to a long-term, purposeful, multi-dimensional, and fundamental transformative process that involves technological, social, institutional, and economic changes in the stormwater management system [10,11].

The multi-level perspective (MLP) serves as the core analytical framework for studying socio-technical transition dynamics. It views transitions as a non-linear process that results from the interplay of developments at three functional levels: landscape, regime, and niche [12]. Landscape usually refers to various external trends (for example, demography, macroeconomy, political culture, societal concerns) and shocks (for example, wars, crisis, accidents) that affect transition. Niches are protective spaces where innovative activity can take place. The regime corresponds to the incumbent system, encompassing the dominant rules and practices that guide activities in particular directions. Based upon MLP, different transition pathways can be outlined, for example according to the timing and the nature of the three levels' interaction [13]. Historical transition pathways that trace the emergence of sustainable regimes could inspire ideas of transition management [14], meaning that sustainability transitions could be facilitated by purposive intervention (governance).

Reflecting its origins within European-capitalist social contexts, socio-technical transition theories have been applied to examine numerous western developed cases from energy, agro-food, transportation, and the water sector. Several researchers have used a socio-technical approach to provide insights on 'what characterizes and shapes the SSWM transition'. Some focus on the competition processes between technology choices. For example, four barriers that affect the adoption of alternative techniques in French stormwater management (difficulty in behavior adaptation, unclear advantages over traditional sewerage method, legal constraints, and the reluctant manufacturers) were examined using transition theory [15]. Some highlight the role of actors, for example, the importance of actor-networks launched by a small group of actors across various sectors and the networked bridging organizations throughout the transition of Melbourne stormwater quality management [16]. Others have explored the timing and types of actions. Ref. [17] focused on the link between technical attributes, governance and transition stage within a Brussels case study, concluding that soft actions such as manuals and legislation prevailed in the early stage of transition, while decentralized processes and collaboration between formal and informal networks were important for the diffusion of later actions. The development of transition pathways allows comparison between transitions with different contexts. For example, the comparison of WSUD development of Netherlands and Australia concluded that both countries have similar transformation pathways followed by dealignment and realignment [18].

In China, the application of socio-technical theories is mainly limited to the energy and transport fields. A handful of exceptions in the water management sector exist: Ref. [19] used the MLP to understand the dynamics and forces that can induce a leapfrogging development of wastewater treatment; the endogenous (regional) and exogenous (international) innovation processes in early transition to on-site water recycling in Chinese cities [20,21] focus on the water regime transition of the social-ecological system at the county level, whereas the transition process is deemed to be driven by landscape pressure and internal

instability within the regime but ignoring the effect of niche innovations. To the best of our knowledge, there is no such published literature that specifically focuses on the SSWM transition process although numerous studies have acknowledged that the Chinese urban water and stormwater management is experiencing transition through the implementation of SCI [22–27]. Most of these studies focused on recent policy changes and practices or identified opportunities and constraints of taking up the new initiative in a general approach.

It is still unclear to what extent western-centric transition theories describe the transition being experienced in China and, if they do, what does it tell us about the stormwater transition known as the SCI? There is a need to assess the cross-cultural robustness of transition theories as transitions differ by nature, scope, and driving forces [28]. Moreover, how is the case in China different from that of other countries, and importantly, does this lead to unique challenges?

This article applies a socio-technical analytical approach to historically reconstruct stormwater management trajectories in the United States and China. The U.S. is selected because it is well-studied in terms of SSWM and is the origin of LID. It is also arguably a major international source of experience for the SCI. These two countries have similar scales of economies, vast territories, and various climates. They also have distinct political, legal, organizational, and cultural characteristics and history. Therefore, the comparison of their stormwater transition process in this study aims to:

1. identify the current stage of the two countries' SSWM transition;
2. identify unique opportunities and challenges for the Chinese SCI;
3. reflect on the feasibility of applying socio-technical transition theory to compare transitions in federal and socialist societies.

## 2. Rational and Approach

This study begins with an overview of the SSWM transition stories of the two countries. A descriptive, country-based approach is adopted. The historical analysis is performed at the country-scale to describe the nationwide transition under different social systems. Drawing on various literature data sources, including policies, regulations, guidance documents, history articles, media reports, and peer-reviewed journal articles, the transition stories are told from the rise of social awareness around the need to modify modern reticulated systems to address growing environmental and social issues. For the U.S., post-World War II is considered the first key node of the transition, when rapid urban expansion and booming industrial sectors greatly increased domestic and industrial effluents, intensified drainage pressure, and resulted in serious water pollution. For China, the timeline was drawn after the founding of the People's Republic of China (1949) when the recovery of urban drainage systems began, alongside a renewed focus on cities and industry. Since it is unrealistic to involve every issue that is relevant to water and environment, only those considered important for stormwater transition are discussed.

A socio-technical transition study using MLP usually needs to assign different elements to the three levels (landscape, regime, and niche). In this study, the regime refers to the stormwater management paradigm at the national level. Examples of elements may include dominant stormwater practices, relevant policy, legislation (rules), guidelines, institutional arrangements, etc. The niche level mainly includes local practice (for example, local innovative policy, local legislation, social activities), and technological innovations. The landscape factors can be represented by slow-changing trends such as urbanization, population growth and mobility, climate change, macroeconomics, macropolitics, social values, culture patterns, and disruptive shocks such as environmental crisis, war, and disasters. For ease of interpretation, the timelines of major changes at the three levels (not fixed) are illustrated in Figures 1 and 2. While it should be noted that these elements are not bound to their level but may evolve to other levels, for example, the local social activities may scale up to large-scale social movements which could be regarded as a landscape pressure to the regime. The boundary between may not be very distinct, for example, the

policy and market can sometimes be regarded as part of the regime and other times as landscape factors. In a word, the assignment of elements is context specific. Therefore, in the text we are not going to separately describe the development at each level (in the way that some transition studies have) but we will examine them in the specific context.

| | Major landscape change | Tangible regime response | Key niche activities |
|---|---|---|---|
| **Predevelopment** | Post war (1945–) prosperity (industry boom, suburbanization, population growth) | The Federal Water Pollution Control Act 1948; Large–scale hydro–engineering construction (since 1930s) | Discontent among public towards environmental degradation; states' actions on quality protection |
| | Environmental movement enlarged from niches (since 1950s) | The Water Quality Act 1965, setup of the Federal Water Pollution Control Administration (1965–1970) | Influential publications (e.g., *Silent Spring*), setup of environmental organizations, public protests |
| | Fire disaster on the Cuyahoga River (1969) | Setup of EPA (1970); The Clean Water Act (1972) | |
| **Take–off** | UN Conference on the Human Environment(1972) | The Endangered Species Act 1973 | Growing public concern on stormwater ecological impact |
| | | National Urban Runoff Program (1979–1983); Amendments to the CWA (1987) | First usage of the term 'LID' in land use planning (1977); first state rule requiring treatment of stormwater, Florida (1982); statewide infiltration standards, Maryland (1984) |
| | Earth Summit 1992 (for climate change and sustainable development | Phase I stormwater regulations (1990); Regulations of TMDL | Dam removal movement gradually began (since 1990s) |
| | | Phase II stormwater regulations (1999); release of LID manual for national use (1999); LID literature review by EPA (2000) | First LID manual, Maryland (1997) |
| **Acceleration** | Failure of the structural flood defence, New Orleans (2009) | Memo: Use of GI in NPDES Permits and Enforcement (2007) | Green Alley Programs, Chicago (2005); calls from scientists and practitioners for consistent, national rule (2009) |
| | | EPA released questionnaires on stormwater practices (since 2010s); revision of MS4 rules (2014); setup of National Municipal Stormwater Alliance (2016) | |

**Figure 1.** The transition trajectory of the U.S. stormwater management.

| | Major landscape change | Tangible regime response | Key niche innovations |
|---|---|---|---|
| **Predevelopment** | Planned economy, *danwei* system | Urban drainage system recovery (since 1950s, suspended from 1958 and restart in 1980s); Large-scale hydro-engineering | |
| | Guanting reservoir pollution (1972); UN Conference on the Human Environment (1972) Reform and opening up (since 1978) | The Water Pollution Prevention and Control Law 1984; The Environmental Protection Law 1989 | |
| | Earth Summit 1992 (for climate change and sustainable development discourse); Pollution incident of Huai River (1994) | The Ninth Five-Year Plan for water pollution prevention and control (since 1996) | Government-civil environment activities (e.g., China Environmental Protection Century trip) |
| | | | Sponge city concept for flood control was first mentioned (2003); local institutional innovation of River Chief |
| | Drinking water crisis in Southern China (2007) | Regional and Watershed Restriction Approvals (2007); Three Red Lines (2011) | Stormwater research burgeoned; First LID demonstration site, Shenzhen (planning since 2011); sponge-like city was proposed on the 2011 Two Sessions |
| **Take-off** | Beijing 712 storm event (2012); the strategic decision of ecological civilization construction (2012) | The strategic decision on the construction of ecological civilization (2012); Sponge city construction was requested by President Xi (2013); Outline of the Integrated Plan for Urban Drainage (Rainfall) and Flood control (2013) | |
| | Conferred legislative power on cities (2015); special provision on public participation in environmental protection (2015) | National technical guidance for sponge city construction (2014); selection of sponge city pilots (2015, 2016); The Strategy of Yangtze River Protection (2016); Assessment Standard for sponge city construction effect (2018) | China Economic Weekly report on urban flood issues of sponge city pilots (2016); public doubts over SCI implementation; Shutdown of dams and hydropower stations, Zhangjiajie (since 2019) |

**Figure 2.** The transition trajectory of the Chinese stormwater management.

Based upon the historical timeline, the transition phases and pathways of the two countries are analyzed. The multi-phase model is often used to identify different transition stages. Four phases were identified: predevelopment, where the regime is relatively stable while some niches may emerge under landscape pressure; take-off phase, where change starts to build up and the regime begins to shift; acceleration phase, in which structural changes occur in a visible way; and stabilization, when the new system reaches equilibrium again [29]. These stages have general descriptions in some of literature [13,29–31], making comparisons requires indicators developed specifically for SSWM transition to make the evaluation criteria consistent (Table 1). The indicators for entering stabilization are some-

what open to debate since the implications for SSWM or sustainability are subjective and dynamic, while such uncertainty does not affect the comparison in this study as transitions in both countries are considered have a long way to reach stabilization (as discussed below).

**Table 1.** Indicators of transition phases.

| Transition Phase | Indicators in This Study |
|---|---|
| Predevelopment | • The stormwater management regime is centered on drainage issues. End-of-pipe and hard engineering logic is dominant.<br>• Niches may emerge by the introduction of innovative technologies, local practices and activities. |
| Take-off | • The regime incorporates some of the innovative niches to reorient its stormwater practices.<br>• Broader objectives of stormwater management come into sight. |
| Acceleration | • Varied actors and stakeholders are involved and a learning and supporting network is shaping between them.<br>• Innovative knowledge and experience is well spread.<br>• The new stormwater regime starts to provide comprehensive benefits across socio-cultural, economic, and ecological sectors beyond stormwater quantity and quality issues. |
| Stabilization | Open to debate |

The typology given by [13] is a common reference tool to identify transition pathways: transformation, reconfiguration, technological substitution, and de/realignment. Under a transformation pathway, underdeveloped niche innovations fail to break through to wider levels whereby actors gradually modify the direction of development trajectories and innovation activities. In reconfiguration, niche innovations are well developed and are being incorporated to trigger subsequent adjustments and change to the regime's basic architecture. During technological substitution, niche innovations have developed to replace the regime; and under de/realignment, landscape pressure creates space within the regime whereby niche innovations co-exist and compete for extended periods until one of them replaces the regime.

## 3. Stormwater Management Trajectories

### 3.1. United States

3.1.1. Predevelopment: Suburbanization and Raising Discontent of Water Pollution

Before WWII, industrialization and urbanization had progressed in the U.S. for more than a century (if regarding the textile industry pioneered in 1790s as the beginning). The new round of urban expansion set off by the post-war prosperity expressed itself differently: revitalized factories relocated to rural areas to reduce costs. Nuclear families (a family group consisting of parents and their children), promoted as the ideal family structure to stimulate the economy, were encouraged to move to suburbs as the proliferation of automobiles enabled further travel distant. As a result, farmland, forest, and undeveloped green space were converted to transportation infrastructure and estates, which contributed to increased surface runoff and pollution.

These land use changes were reflected in the deterioration of water bodies, strengthened the landscape pressure while on the other hand provided unique niches for raising public environmental awareness. When people moved to the suburbs, interests in outdoor recreational activities that closely related to nature grew as one of the romantic fantasies of nuclear family life. Ironically, environmental degradation quickly dispelled such fantasies. Suburban dwellers, mostly upper-middle class individuals, later became the major force of the national environmental organizations and actively participated the massive environ-

mental movement from 1950s to 1970s [32]. The 1950s and 1960s saw a series of influential publications (for example, Silent Spring), the setup of nonprofit organizations (for example, Nature Conservancy), and public protests (for example, against construction of the Echo Park Dam) on environmental protection. Some local officials and communities which faced with significant pollution hazards began to take active measures such as establishing local water boards and financing clean-up programs. It should be noted, however, that this was built on the earlier efforts of conservation movement (1890–1920), the prevalence of preservationism and wilderness since the middle of the 19th century. In other words, it took a rather long process for public perceptions towards their relationship with nature and the environment to realize such transition.

The stormwater regime at the time still focused on drainage and sanitation issues and retained a strong end-of-pipe, hard engineering logic. There was a dominant and pervading belief in science and technology, which made engineers the core urban solver [33]. The construction of large-scale hydro-engineering projects, especially dams since the Great Depression (1930s), went further; these were driven by the demand of economic stimulation and extended political influence [34]. The Federal Water Pollution Control Act 1948 (FWPCA) firstly required a cooperation between federal and state entities to address declining water quality. However, the Act only 'encouraged' pollution control without federal supervision [35]. Federal efforts mainly limited in assisting the construction of large-scale grey infrastructure such as sewers and treatment plants [36]. States retained primary power and responsibility of water pollution management. By 1966, all states had passed some type of water pollution legislation, but enforcement varied greatly [37]. For many state governments, the priority of economic development was still unassailable. Consequently, the federal encouragement and local clean-up efforts could do little to regulate rivers and streams especially those issued with industrial permits under state legislation.

### 3.1.2. Take-Off: Ecology Concern and Stormwater Quality Legislation

The 1969 fire disaster on the Cuyahoga River, Cleveland, which was caused by accumulated industrial pollution from storm water overflows and raw sewage discharges, was a notable shock to the regime, triggering widespread distrust of the ability of state and local governments to effectively manage water quality [38]. It is generally believed that this incident directly led to the establishment of the Environmental Protection Agency (EPA) in 1970 and the revision of FWPCA in 1972 (the Clean Water Act, CWA) [39]. While the 1969 Cuyahoga event was not the first and even not the most serious river-fire disaster, the significant regime change was considered more deeply promoted by the relatively fully developed niche activities and the already destabilized regime.

The 1972 FWPCA explicitly separated point and non-point source pollution, the former became subject to the National Pollutant Discharge Elimination System (NPDES) permit, however, non-point source pollution was still not specifically addressed. Pushed by the prevalent environmental activism that aimed to protect the nation's eco-heritage from extinction, the Endangered Species Act of 1973 was passed. The Act offered a new window for bringing stormwater quality into the regime as bio-indices for quality evaluation were introduced, whereby a higher standard of water quality was required for habitat reservation [40]. Under this Act, a growing number of biologists and environmental groups expressed their concern over the impact of stormwater on ecosystem health. In 1979, EPA launched the National Urban Runoff Program (NURP) to investigate the impact of stormwater on species' habitats, whereby its final report formally determined stormwater as a pollution source. The report highlighted the effectiveness of detention basins, retention ponds, and wetlands on capturing contaminants from stormwater runoff. These findings in turn spurred increasing interests on nature-based solutions to stormwater management. During the same period, several states and municipalities started formulating stormwater ordinances demanding on-site storage and detention (for example, Florida, Pennsylvania). Maryland took the lead by creating a statewide infiltration program during the mid 1980s, which became the origin of LID.

Building upon the results of NURP, the next decade saw significant changes in the U.S. stormwater regime. The amendments of CWA in 1987, section 402(p) in particular, brought stormwater under the NPDES permit (National Research Council 2009). States were required to develop and implement nonpoint pollution management programs. Combined sewer overflow (CSO) was identified as a point source, whereby its control was guided under the National Combined Sewer Overflow Control Strategy in 1989 and its revised version in 1994. In response to the revised CWA, the Municipal Separate Storm Sewer System (MS4) program was launched in two phases (Phase I, 1990-; Phase II, 1999-) to require a NPDES permit for stormwater discharge and submission of a stormwater management or pollution prevention plan.

Spaces for niches to grow were also intentionally created through these regime changes. In 1992, the Total Maximum Daily Load (TMDL) limit describing the maximum pollutant that a water body can receive was incorporated into CWA. Most states were responsible for developing TMDLs and submitting them to EPA for approval. For water that did not meet the minimum criteria, cleanup plans that were essentially watershed-based were required. This promoted the establishment of collaborative watershed groups (for example, partners of Chesapeake Bay program). Moreover, public participation was mandated in specific provisions: the Phase II rule defined six elements as minimum control measures that contributed to successful stormwater programs: public education and outreach, public involvement/participation, illicit discharge detection and elimination, construction site runoff control, post construction runoff control, and pollution prevention/good housekeeping [41].

### 3.1.3. Nowadays: Exploration of Multiple Benefits from Stormwater Management

While the MS4 programs did not strictly require the implementation of specific runoff control measures, it did create momentum for LID adoption. In order to help local governments and developers comply with stormwater legislation and regulations, the EPA along with organizations such as the Natural Resources Defense Council and the LID center made a concerted effort to spread the stormwater experience of Maryland. The diffusion of LID was accelerated via the release of manuals that guided its national application [42], case reports containing evidence of its cost-effectiveness [43,44], the establishment of websites that assembled policies and training resources of sustainable stormwater issues (for example, http://water.epa.gov (accessed on 12 June 2020)), and databases that offered precedent design and performance of stormwater devices (for example, the Best Management Practices database). LID for flood mitigation was also emphasized, especially after the flood defense failure in New Orleans in 2005, which again highlighted the deficiency of hard engineering flood control approaches and the necessity for transition [45]. The dismantling of dams since the end of 1990s also signaled the erosion of the past hard-engineering regime.

Green infrastructure (GI), a concept that originated from landscape planning, shares similar logic with LID in mimicking natural hydrological processes. The popular Green Alley programs in several U.S. cities (for example, Chicago and Los Angeles), although initially stormwater-focused, were acclaimed to provide ecological and cultural services [46]. A memo released by EPA and the Energy Independence and Security Act of 2007 endorsed GI as a wet weather infrastructure solution [47]. Federal agencies were required to reduce stormwater runoff via LID and GI from federal projects [48].

Despite LID having been increasingly used in the U.S. since the 1990s and generally positive feedback has been received from previous practice, LID adoption and implementation at the national level is not as expected [48–51]. The enforcement and outcomes of stormwater programs are also highly variable [52]. Multiple reasons have been identified such as technical infeasibility, lack of financial support, uncertainty of LID cost-effectiveness, challenges in operation and maintenance, conflict of stormwater regulations and other local ordinances, etc. While some of these challenges might be common to all countries, the large degree of discretion left for states to have their own governing system (metric, review processes, institutions setup, etc.) and legal codes is the most distinct characteristic of the

U.S. system which may be beneficial for local-based decision-making but complicates policy transfer between cities and states, making both tracking compliance to federal regulations difficult. Although there have been strong calls for consistent nation-wide rules [52] and commitments by EPA to enact them, in 2014, the EPA deferred on such actions, insisting on leveraging existing requirements and playing a supporting role in the implementation of local stormwater programs.

The regime actors decided to move to a more cooperative manner instead of a more top-down approach. Since 2010, the EPA started to distribute thousands of questionnaires to MS4s, NPDES permitting authorities, and developers, to collect specific information about their stormwater practices, oversight, and finance issues to inform how could EPA improve its stormwater program [53]. The Nonpoint Source Outreach Toolbox has been continuously updated in recent decades in order to assist state and local agencies and other organizations with educating the public on nonpoint source pollution or stormwater runoff [54]. In 2016, the National Municipal Stormwater Alliance was established to represent MS4 permittees at the national level, providing a unified voice when working with the EPA, states, regional municipalities and other stormwater organizations. In the same year, the EPA issued a revision of the MS4 rules, which resulted in more flexibility for permitting authorities to issue and administer small MS4 permits. A draft guide, toolkit, and technical assistance was released to promote comprehensive, community-wide planning approaches to manage stormwater [55].

### 3.2. China

### 3.2.1. New China: Restoration of Drainage Infrastructure, Rising of Technocrats, and Restricted Exposure to Nature

Stormwater drainage has a long history in China. The well-known Fushou drainage system built in the Northern Song dynasty (960–1127) includes both combined drainage and storage infrastructures, with one of the sections still functioning today [56]. People in the agricultural society viewed domestic sewage as precious fertilizer which usually would not be discharged into stormwater sewers [57]. However, due to a century of war and drastic social changes since 1840, the Chinese urban economy and infrastructure were badly damaged. Therefore, one of the important tasks of urban planning after the founding of the People's Republic of China was the basic sanitation remediation of major cities through sewer construction and retrofitting, as well as filling up significantly polluted river channels for sanitation and urban development [58]. However, this endeavor was soon halted by the Great Leap period (1958–1960) and the Great Cultural Revolution (1966–1976). Following this period saw the emergence of technocrats as social order returned and the revolutionary veterans stepped down from decision-making positions. Drainage recovery work rapidly resumed under the supervision of these technocrats. Given the lack of funding and mature technology in China at the time, most of the drainage systems were built as combined sewers. The design standards were far below that of the U.S. For example, the recurrence interval of most drainage systems in Chinese cities were usually in a range from 0.5 to 5 years (5 to 10 years in the U.S.).

As the economy steadily grew, China's population rose rapidly from 0.54 billion in 1949 to 0.83 billion in 1970. Following the similar route, increased domestic and industrial effluents led to the degradation of water quality in China's aquatic environments. Meanwhile, technocrats kept enlarging their influence due to the continued to prioritize industrialization and economic development of macro policy. Similar to the U.S. hydro-engineering construction boom since the 1930s, multiple plans for large dams and hydropower stations were implemented for flood control, cheaper energy, and economic stimulation. Waterways were modified or filled for further development. This development paradigm was underpinned by the once-popular political slogan 'rendingshengtian', which means 'man can conquer nature'. The 'antagonistic' mindset together with additional impacts of dam construction (for example, population migration) were pushing people away from water and environment both physically and cognitively.

While compared with the U.S. case which finally developed strong enough bottom force to affect the regime, there was little sign of influential niche activities in China at the time. The highly centralized planned economy and the dominant urban society organization approach, as well as the *danwei* (work unit) system that attach people's residence to their workplace are considered as the two major landscape factors for this lack. The Five-year Plan, developed to set national development goals under the planned economy, was over-reliant on pure administrative means, which greatly inhibited niche-level activities. Its profound influence on subsequent environmental management remained for a long time even after it changed to a socialist market economy in the 1980s. The *danwei* is typically enclosed by walls, which implies security and has a unique social identity for Chinese people [59]. Such values contrast with western desire for outdoors life (recall the nuclear family fantasies). Most social and recreational activities took place in the *danwei*, which restricted the exposure of city dwellers to nature and therefore minimized public concerns towards degraded environmental water quality.

3.2.2. Evolving Water Quality Management: Tightening Regulation and Point Pollution Source Focused

A major perceived shock that opened a window for the water quality regime was the 1972 pollution event of the Guanting Reservoir, Beijing, which killed numerous fish and threatened the health of locals [60]. This event coincided with the first instance of Chinese participation in the 1972 United Nations Conference on the Human Environment, which was the first world conference to make the environment a major issue. In response to this pollution event, a series of local measures were immediately adopted to deal with the reservoir issue, whereas little had been addressed for pollution in other areas due to the near absence of legal and institutional systems of environment protection.

Although the global environmental movement was not as influential in China as it was in the western world, certain worldviews were still assimilated by top leaders, which brought new opportunities for change. The first act on water pollution, the Water Pollution Prevention and Control Law was passed in 1984. The Ninth Five-Year Plan initiated in 1996 aimed to improve the water quality of major rivers and lakes by 2000. The 1990s saw some government-led environmental campaigns (for example, the China Environmental Protection Century demonstration) and the formation of some early environmental groups.

Entering into the 21st century, urbanization in China has already developed to an astonishing scale. The urban population has tripled in the past three decades, driven by factors such as migration policies, rural-urban disparity, and land development for urban use [61]. On the one hand, the centralized managed water system in cities allows people to have access to stable water supply and sanitation. While on the other, such rapid growth has been accompanied by the substantial pollution risk of drinking water resources. The serious eutrophication of major freshwater resources in Southern China, especially the event known as the Wuxi water crisis, served as another catalyst that urged more stringent instruments to be adopted.

In 2007, the Regional and Watershed Restriction Approvals was implemented to limit the number of emerging polluting enterprises from the central government level [62]. The Three Red Lines set for water consumption, water use efficiency, and water environment restoration in 2011 is regarded as the most rigorous regulatory regime for water resources management. Some local institutional innovations became the regime. The River Chief System, first experimented in Changxing county, Zhejiang Province in 2003, was incorporated into law in 2017 to strengthen the responsibility of individual government officials on river health management. These instruments now play a key role in preventing local governments from sacrificing environmental interests, while also consolidating the reliance of top-down administration. However, stormwater was still marginalized at this stage and seldom involved in these point source-focused actions.

### 3.2.3. Emerging Niches and the Proposal of Sponge City Initiative

As the water quality regime was experiencing incremental transformation, research interest in stormwater sector gradually started to accumulate, which was, to a great extent, attributed to the global flow of relevant knowledge. In 2003, Yu and Li [63], a group of researchers in landscape design who were under the great influence of western landscape perspective proposed the idea of the sponge city and drew an analogy between wetlands and sponges, emphasizing its function in absorbing excess water. They held the view that current planning and development ignored the important role of nature in regulating river flow and flood. Some demonstration sites designed by them were heralded as the predecessor of sponge city experiments (for example, the Qunli wetland park in Harbin). Scientists from Beijing University of Civil Engineering and Architecture led by Che and Li specifically focused on LID utilization for stormwater management [64–66]. The number of studies on runoff quality also grew during this period [67–69].

The Chinese market once offered niches when a number of development projects labeled as 'eco-friendly', 'green', and 'low carbon' cropped up in some cities from 2000s, whereas stormwater retention and water recycling were promoted as one of the 'selling points'. Local governments were supportive because it seemed to be an appealing way to improve city image and attract foreign investment on environmental and ecological grounds. However, without sufficient scientific proof and institutional support, these projects evolved into tools for financial gain rather than environmental protection, with some of these projects being abandoned mid-development (for example, Dongtan Eco-city, Shanghai) [70,71]. A few recognized successful cases do exist, usually developed later around the 2010s in the form of urban planning, such as the early planning of Guangming District in Shenzhen which first introduced the LID concept.

Compared with more invisible stormwater quality, the general public in China is no doubt more familiar with urban flooding issues. Floods in cities across the country had become commonplace. According to People's Daily, 62% of 351 cities in China experienced waterlogging (neilao) from 2008 to 2010 [72]. People started to express their distress over waterlogging through a popular sarcastic joke: 'watch the sea in the city' (chengshikanhai). Under the heat of the complaining, in the 2011 Two Sessions, one of the National People Congress deputies proposed to build 'sponge-like' cities using greenbelts and sloped pavements as a solution to urban flooding. The next year, the Beijing storm disaster on 12 July resulted in at least 79 deaths and about 11.64 billion *yuan* economic loss (approx $1.86 billion USD). These significant casualties and economic losses generated considerable public discontentment towards conventional flood management and lead to growing appeals from scholars to implement stormwater management approaches to improve city resilience to climatic variability. At the 2013 Central Working Conference of Urbanization, President Xi formally put forward the request for nationwide sponge city construction. Although the Beijing storm disaster on July 12th was clearly an important SSWM transition node, it is important to appreciate the other contextual factors which formed the transition, such as the experience from previous niche activities, profession's efforts, changes in public perception, etc.

Urgent demand for flood control is clearly a dominant driver, however, the motivation from ecological sustainability promotion cannot be ignored. Notably, at the end of the same year of the Beijing storm shock, the 18th Communist Party of China National Congress made the strategic decision of 'vigorously promoting the construction of ecological civilization'. Sponge city is regarded as the practice of ecological civilization in urban water management. The Strategy of Yangtze River Protection and the wave of ecological protection in the Yellow River basin established by the central government in 2016 and 2019, respectively, were not only about protecting the river itself, but sought to regulate urban development in cities along the two major rivers. Many projects under the two programs were integrated with local sponge city construction. More comprehensive objectives of sponge city have also been developed in plans and guidance. The Assessment Standard for Sponge City Construction Effect released in 2018 involves requirements for runoff reduction, CSO

control, ecology, groundwater, urban heat island, etc. [73]. After reviewing a number of local sponge city plans, the authors concluded that except for the requirement of runoff quantity control, the common controlling indices (kongzhixingzhibiao, in some cities, also referred as mandatory indices) are riparian ecosystem restoration, water surface ratio, and nonpoint source pollution control. The utilization rate of stormwater may also be a controlling index in cities suffering from severe water scarcity due to pollution or drought, for example, in Hefei and Guiyang.

### 3.2.4. Insufficient Follow-Up Knowledge and Experience Diffusion

Immediately after this conference, a national technical guidance for sponge city construction was published to advise the usage of LID infrastructure to deal with stormwater problems [2]. The Chinese-style pilot projects played a unique role in promoting local stormwater experiments at this early stage. In 2015 and 2016, 30 cities were selected in two batches as sponge city pilots by the ministries of State Council, which implied that they could obtain special financial support and attention on this issue from the central government. All local governments are required to complete the drafting of special plans for sponge cities and devise an integrated plan of urban drainage and flood control [74]. The speed of top-down execution was astonishing (at least on paper), whereby at the end of 2017, almost all major cities have incorporated sponge city in their urban planning documents. The central government also sought to strike a new balance of power and responsibility with local governments. The 2015 amendments to the Legislation Law conferred limited legislative power on all cities, which brought more local wisdom and innovations. For example, Hebi (one of the 30 pilot cities) formulated its first local laws and regulation on the Construction of Circular Economy and Ecological City, which includes a specific chapter for sponge city.

Nevertheless, when the China Economic Weekly [75] reported that 19 of the 30 pilots were still experiencing severe urban flood or waterlogging in rainy seasons, including Beijing, Tianjin, Wuhan, etc., there was widespread skepticism towards the effectiveness of the sponge city solution [76], and criticism among researchers on SCI implementation such as lack of a fully understanding of sponge city concept [77], the trend of 'one model fits all' [78]. The Ministry of Housing and Urban-Rural Development (MHURD) responded by saying that 'sponge cities cannot be built overnight and its construction is still accelerating'. Besides, although SCI was proposed with a mixed of policy goals, the dominant expected effect from sponge city programs was significant improvements in flooding prevention. The timing of the regime shift and its close association with the flood shock created a misleading context that sponge city was only designed for flooding prevention. As a result, public perceptions towards sponge cities can be easily swayed by unsatisfactory outcomes when it rains heavily.

Yet despite these more incidental factors, the institutional arrangements that left little space for public engagement and the lack of cooperative and learning networks might be the root causes of these despairing public perceptions. Although recently the amendment of Environmental Protection Law regulates special provision on public participation in environmental protection [79], which normalized procedures such as public hearings and opinion collection in decision-making, has opened a window for public engagement. So far, citizens are more likely to be passive recipients that are only informed after decisions have been made [80]. In addition, government-led policy propaganda was almost the only source for people to understand the initiative. The failure of central government to diffuse comprehensive knowledge on the implication of sponge cities compounded negative perceptions especially because the public is rarely aware of the benefits of sponge cities in other sectors. SSWM transition can be stymied because these misunderstandings could result in unwillingness to be involved.

### 3.2.5. Still Firm Incumbent Hard Engineering Approach

LID was not the only component of sponge city projects. In many cities, the construction of large rainwater storage tanks, pumping stations, traditional drainage systems, and deep tunnels projects are listed in local sponge city plans and received substantial investment (for example, Shanghai, Guangzhou, and Wuhan) [77]. They served as a remedy for low-standard drainage systems constructed prior to the sponge city. LID or GI infrastructure is encouraged but not required or regulated. When flood control is emphasized as the major stormwater management goal, it may take a long time for LID to become equally competitive as traditional technologies, unless other innovations or disruptive changes occur. Although recent years also saw the shut-down of some dams and hydropower stations, exemplified by the case in Zhangjiajie, a national park in Hunan province to protect giant salamander habitat, hard engineering approaches to flood management were still held in strong faith at the meso level.

## 4. Discussion

### 4.1. Theory Validation

Stormwater management transition in China and the U.S. is in accord with the general multi-level dynamic. At the landscape level, the two countries share similar trends in urban expansion and development, economic development, population growth, climate change, and supranational activities over environmental protection and sustainable development. Flooding and water pollution events play an important role in raising awareness of the necessity of stormwater management transition, as it is usually an underlying social problem that people would not notice until the system fails. These contribute to the destabilization of the incumbent regime and provide momentum for niche activities to grow. Landscape changes do not always lead to positive transitions. For example, compared to the post-war relocation and family structure change in the U.S., the highly centralized production and activity space after liberation in China during the 1950s and 1960s inhibited niche opportunities, especially for the general public. From this perspective, we assert that landscape is a crucial functional level especially in SSWM transition research because its influence ultimately unfolds in the reshaping process of people's perception of urban space and their relationship with the water environment, according to which the transition process can be either hindered or proceeded. This could be the meeting ground between socio-ecological and socio-technical research considering that the former focuses more on the society-nature interaction [81].

In both countries, local experiments on LID or sponge city preceded national actions, although their effect and impact on the regime vary. In the U.S., the driving force from the bottom was prominent due to multiple actors and various niche activities involved in the process, especially social groups and the general public. They served as a strong catalyst of internal niche innovations and then expanded their voice to the macro level, urging the government to make the change. The clear bottom-up characteristics are similar to many western countries that experienced environmental movements from the 1970s. Meanwhile, the regime actions also led to changes on the ground, such as the NURP investigation conducted by EPA that inspired localities stormwater impacts on ecology system and research interests over LID, the MS4 programs that drove innovative management strategies and technologies, the multiple organizations that brought different groups into the conversation and so on.

In the Chinese case, governments and techno-professions were dominant niche actors, whereas the public had little awareness and access to stormwater issues. The incentives of early stormwater experiments were more of an economic gain for developers and local governments compared with improving environmental outcomes. The need for transition was not completely recognized by the regime and neither was the value of local innovations. The opening up of the opportunity window of transition [82], which depends on when the institutional context permits niche innovations to be introduced, was largely delayed until the Beijing flood disaster in 2012. Consequently, the niche development was

relatively underdeveloped when the SCI started to be promoted in China compared with LID implementation in the U.S. Lack of additional pressure from grassroots movements made both niche and regime changes less radical. However, these changes were still important in reshaping the institutional context and providing valuable theoretical and practical experience for the proposal of SCI.

Transition is a process of the long-term modulation between the three levels. The significance of the activities at the macro and the micro level should be paid with special attention, rather than just focusing on the resultant regime changes. It may be easy to describe the U.S. and Chinese transition as 'bottom-up', where micro niches cluster and influence the incumbent regime, and 'top-down', where a large and rapid change in landscape leads to regime change. However, these terms should be used with caution because they only describe the surface trigger mechanisms and may in fact hide the idea that the transition is a long-term process with influence from all three levels. There is, of course, a possibility that the transition is the result of being squeezed between top-down and bottom-up influences, which is most likely the case in both countries.

*4.2. Transition Pathway*

Before the enactment of stormwater quality legislation in the U.S. and the proposal of SCI in China, the water quality regime had experienced gradual, cumulative adjustments from loose regulations to strict effluent discharge restriction, from local affairs to federal or national government supervision, and reorientations of the development path from 'all for economy' to 'ecology matters'. These changes form necessary (but not necessarily sufficient) conditions for transition. Although stormwater management was generally marginalized in urban issues, its concern seems to be only a matter of time as the result of increasing demand for water quality and ecology conditions. However, the length of that time, and the effect of the quality and ecology concern on stormwater transition are different in the two countries.

For the U.S., the provision of ecosystem health is the major driving force of stormwater management transition. The stormwater sector underwent restructuring major legislative changes, followed by the centralization of power in the federal government. The emerging technologies initially adopted to solve locally specific problems were then expanded to a broader geographical range and usage. If the changes in water quality management are considered as moderate pressure to the stormwater regime, then it is fair to say that the transition in the U.S. began with a transformation pathway and followed with observable signs of reconfiguration.

In China, the demand for urban flood control largely drove the proposal of SCI. Admittedly, the Beijing flood disaster is a shock to the regime and the proposal of SCI is a breakthrough of stormwater policies. However, the regime did not experience a shift as radical as that after take-off in the U.S., rather it changed in a more moderate way under the mixture of policy goals and processes which interacted between SCI and other sustainability programs. There are no visible changes to the basic structure of the regime. SCI is still implemented through a state-planning system that is far from complete. Not only does the existing permit system of pollution discharge only target the point source, the urban planning regulations also do not require any order/priority between stormwater discharge and land use [83]. Apart from the duty of municipalities to ensure basic drainage and sanitation conditions (the feature of the incumbent regime), stormwater quality management is essentially non-mandatory.

Differentiated from other technical innovation-based transitions, LID is more of an artificial approach to simulate natural ways of regulating water. There is no doubt that the technical innovations at the niche level are increasingly more developed, at least sufficient to be alternatives for regime actors. However, they are still unable to entirely replace existing drainage systems and may have to co-exist with the previous regime for a long time. The transition itself is therefore not a complete technological substitution in both

countries. In addition, the possibility of other particular technologies being fully developed to substitute others should not be excluded.

LID is considered less competitive in China, largely determined by the functional features of this type of technology and the institutional settings that encourage them. While LID is especially useful for reducing pollution from first-flush and mitigating the runoff directly entering into drainage systems, it could be inadequate when dealing with flooding problems in extreme climate events. Compared with the regulatory restrictions on stormwater quality in the U.S. which endow more motivation for LID adoption, the Chinese stormwater management regime is largely centered around quantity. Therefore, traditional stormwater management approaches are still privileged.

*4.3. Transition Phases*

According to the defined phase border, the predevelopment and take-off phase of the SSWM transition in the U.S. and China can be demarcated around the revision of CWA in 1987 and the sponge city proposal in 2013, respectively. These two actions mark the reorientation of stormwater practices, the beginning of extending stormwater management to more than previous hard-engineering, end-of-pipe, single-purpose drainage. The focus of this section is on whether they have entered into or how close they are to the acceleration phase used in transition theory.

In the U.S., the new regime starts to provide multiple benefits beyond stormwater treatment and there are many indications of a general desire for further changes, either in the promotion of LID or in stormwater governance improvement. Given the experience of local governments, experts, social groups, and the general public in expanding their influence on the other two levels, actors at different levels have relatively strong connections. A support network promoting collective learning is shaping. Under the common effort of the government, research institutions, non-government organizations, industry groups, etc., the adoption of various approaches of knowledge and experience dissemination for LID promotion appears in time, helping smooth the transition from take-off to acceleration. Cross-sector organizations established in recent years bring varied actors together towards the same sustainable goal. The engrained difficulty mainly lies in achieving state/federal balance and policy transfer between different regions to ensure consistent enforcement and outcomes, which remains a core challenge for many environmental governance problems within the federal system.

The Chinese transition history show a relatively limited institutional space for niches to grow. In particular, the general public had less opportunities to participate. A few early social movements to address point source pollution were under government leadership. Although SCI has been successfully proposed in a relatively top-down approach, the negative impacts of lacking multi-stakeholder participation may not become apparent until it hinders the following implementation of the program and the embedding of the emergent stormwater policy. For example, installing LID devices to private properties such as green roofs or rainwater tanks, local facilities maintenance, and moreover, getting financial support from the public could face strong resistance (one suggested funding source is public-private partnership, which proves to be almost failed). There is also a salient vacancy on the follow-up knowledge and experience diffusion after nearly nine-year SCI implementation. Despite several case studies and the release of local guidance, few reviews based on case studies, manuals, or databases that can be influential across varied regions and audience types have been produced. There is a lack of platforms or intermediary organizations that could promote communication, coordination, and cooperation between different institutions, sectors, and regions. This is especially important considering that many cities have little history of relevant knowledge and experience.

This evidence suggests that the SSWM transition in the U.S. is closer to acceleration while that of China is currently at the early phase of take-off. Nevertheless, the Chinese transition history along with its unique socio-institutional system holds its own opportunities. Compared with that in the U.S., the state-planning mechanism may benefit stormwater

plans developed with more comprehensive goals and allow the integration of stormwater management and the development of other areas such as economic development. Policy transfer may face fewer obstacles because the basic political, legal, and institutional systems of most regions in China are highly consistent. The Chinese central government has more centralized power and experience in managing areas with geographical and cultural diversity. Local areas have a high compliance with state decisions, whereas the U.S. states may have varied responses to the federal government decisions. However, such an advantage could only be realized with a full recognition of local customs and practical knowledge [84] otherwise there might present the risk of 'one model fits all' and its shortcomings.

## 5. Conclusions

This study reveals the different features of the transition toward Sustainable Stormwater Management in the U.S. and China. This was conducted in part by applying the multilevel perspective model which showed the requirement for activities at all three levels (landscape, regime, and niche), no matter which social system to which it was applied. The U.S. transition is marked by relatively developed niche activities and strong links between niche and regime and landscape. The civil society serves as a significant actor. In contrast, niches are less developed in China and the governance and politics of the transition do not explicitly incorporate the public as stakeholders. When we considered LID as the focused innovative technology, the U.S. transition pathway shows a more apparent trend of dealignment and realignment. LID is now symbiotic to the regime rather than a substitution in both countries. Based upon the multi-phase model, the U.S. transition is deemed closer to the acceleration phase while the Chinese one is at the early stage of take-off.

The purpose of this study is not to provide a specific answer, but to provide a new way of thinking. The authors believe that the results of the study play a reminder or warning role for stormwater practitioners: the process of seeking the solution to stormwater management problems, such as obvious preference for grey engineering approach, lack of public participation and investment, etc., should not be separated from studying the transition history of the specific social system. Since these problems are formed under the long-standing influence of multi-level elements interaction. Moreover, when brainstorming the more sustainable way, the ideas should not be limited in fields of explicit stormwater engineering area and should attempt to utilize the current landscape and niche characteristics as new niches. Measures that only hit one aspect of one single level are unable to vacillate current regimes.

However, caution and critical reflection are needed when interpreting the comparison since the conclusion that one country is ahead of another may lead to an overly simplistic heuristic that the latecomer needs to facilitate (the same) changes to converge on the earlier adopter. The transition pathway is full of variability, uncertainty, and complexity and is place- and context-based. Essentially, the comparison does not uncover (and in fact, tends to cloak) an important question: is it necessary or possible for a socialist, central-planning society to learn from the route of a capitalist and more decentralized society (and, if so, to what extent and in what way)? It is unclear whether the current transition pathway will lead the U.S. stormwater management toward sustainability. It is also too early to say whether the stormwater quality-led transition with niche-driven radical changes is the best way to approach the environmental crisis in China. From this perspective, applying western-centric transition theory to such a comparison tends to imply that the capitalist transition is the way forward. This may impede countries with a distinct social system (especially eastern, developing, and socialist) from exploring alternative and more environmentally focused development pathways.

**Author Contributions:** Literature review and historical analysis, Y.Z.; writing—original draft preparation, Y.Z.; writing—review and editing, M.P. and S.T.; supervision, S.T. All authors have read and agreed to the published version of the manuscript.

**Funding:** This research was funded by China Scholarship Council (201908420296).

**Institutional Review Board Statement:** Not applicable.

**Informed Consent Statement:** Not applicable.

**Data Availability Statement:** Not applicable.

**Conflicts of Interest:** The authors declare no conflict of interest. The funders had no role in the design of the study; in the collection, analyses, or interpretation of data; in the writing of the manuscript, or in the decision to publish the results.

## Abbreviations

| | |
|---|---|
| SSWM | sustainable stormwater management |
| LID | Low Impact Development |
| SCI | Sponge City Initiative |
| MLP | multi-level perspective |
| WWII | World War II |
| FWPCA | Federal Water Pollution Control Act |
| EPA | Environmental Protection Agency |
| NPDES | National Pollutant Discharge Elimination System |
| NURP | National Urban Runoff Program |
| CSO | Combined sewer overflow |
| MS4 | Municipal Separate Storm Sewer System |
| TMDL | Total Maximum Daily Load |
| CWA | Clean Water Act |
| NRDC | Natural Resources Defense Council |
| GI | Green infrastructure |
| MHURD | Ministry of Housing and Urban-Rural Development |

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
