# Peer review of "Comparison of the Transition to More Sustainable Stormwater Management in China and the USA"

_water, doi:10.3390/w14121960_

Round 1

Reviewer 1 Report

Climate change is being experienced all over the world. It causes droughts to occur alternately with periods of intense precipitation resulting in flash floods.  On the other hand, the growth of urbanization contributes to the deterioration of water quality. In this context, the solution is to retain a large amount of rainwater and use it in a timely manner. This approach can be implemented on the one hand by using the already existing infrastructure and urban development in accordance with a comprehensive approach that allows for rainwater retention and improvement of its quality and applies to both natural and build-up areas. The article submitted for review is a part of the above mentioned topic. The authors have made a comparative analysis of the transition dynamics and changes in approach to stormwater management in the US and China, taking into account 3 aspects: landscape, regime and niche. They performed their analysis based on various literature data sources, including policies, regulations, guidance documents, history articles, media reports, and articles. In total, they cited 84 pieces of literature, more than half of them from the last 10 years. The work is very clear - the authors have made an introduction to the subject, specified the purpose, discussed the adopted methodology of analysis. The most valuable are chapters 3 and 4, in which they made a thorough analysis of changes in both countries and an interesting discussion. The paper ends with conclusions. The content presented in the paper is very clear to a potential reader. However, I suggest to the authors that due to the use of many abbreviations in the paper, they should list them at the beginning of the paper in the form of a table to make the presented content easier to read. There is also a lack of explanations for some abbreviations introduced.

Author Response

Response to Reviewer 1 Comments

Point 1: The authors have made a comparative analysis of the transition dynamics and changes in approach to stormwater management in the US and China, taking into account 3 aspects: landscape, regime and niche.

Response 1: Thank you for reading the paper. It is nice to see that the message was clear and the paper was understood.

Point 2: They performed their analysis based on various literature data sources, including policies, regulations, guidance documents, history articles, media reports, and articles. In total, they cited 84 pieces of literature, more than half of them from the last 10 years. The work is very clear - the authors have made an introduction to the subject, specified the purpose, discussed the adopted methodology of analysis. The most valuable are chapters 3 and 4, in which they made a thorough analysis of changes in both countries and an interesting discussion. The content presented in the paper is very clear to a potential reader.

Response 2: Thank you for the kind comments. We are glad the paper is clearly written and well cited. And we agree that chapters 3 and 4 are the meat of the paper and where the most interesting part of the work can be found.

Point 3: I suggest to the authors that due to the use of many abbreviations in the paper, they should list them at the beginning of the paper in the form of a table to make the presented content easier to read. There is also a lack of explanations for some abbreviations introduced.

Response 3: This is a great suggestion. The abbreviations have been tabulated for ease of reference. If it is allowed in the Journal please include the attached table as supplementary information at the end as it is the usual format for additional material. The table will address the reviewers’ comments and explain these abbreviations. We have also deleted some unnecessary abbreviations in the text as they only appear once.

Reviewer 2 Report

General Comments:

Dear authors, I congratulate you on a very well-written and structured manuscript. I really enjoyed reading it. I think your work is of great interest to people working in sustainable Stormwater Management even those outside academia. Related to the potential readers of your paper, I think that your manuscript is lacking clean statements on the potential applications and the usefulness of the results that were obtained in this research. Please have in mind the potential broad spectrum of readers eager to read your recommendations based on your findings. Besides this, I recommend focusing more attention on some key cities (particularly in the USA as the text already provided some specific context for some Chinese cities) that can serve as examples of the transition. Finally, are the USA and China the most progressive countries in sustainable Stormwater Management? Are there other cases worth to be mentioned and to some extent discussed? A discussion on why to compare the USA and China is already provided in the manuscript but what is lacking is a critical discussion on how these two countries fit in the global spectrum.

Specific comments:

  • Page 11 - Lines450: Not clear what Beijing 712 storm is.
  •  

Author Response

Response to Reviewer 2 Comments

Point 1: I congratulate you on a very well-written and structured manuscript. I really enjoyed reading it. I think your work is of great interest to people working in sustainable Stormwater Management.

Response 1: Thank you for the kind words. We hope this paper will form a well cited addition to the journal.

Point 2: Related to the potential readers of your paper, I think that your manuscript is lacking clean statements on the potential applications and the usefulness of the results that were obtained in this research. Please have in mind the potential broad spectrum of readers eager to read your recommendations based on your findings.

Response 2: Thank you for your advice and this is an interesting observation. Indeed, we expect the target readers of this paper would be stormwater practitioners that are well versed in the field and so they could interpret the results meaningfully. The usefulness of the results are already presented in the introduction and discussion sections, while we also add a brief clarification in the conclusion part on Page 17 Line 701-Line 711 to make it clearer. 

Point 3: I recommend focusing more attention on some key cities (particularly in the USA as the text already provided some specific context for some Chinese cities) that can serve as examples of the transition.

Response 3: This is an interesting idea. We have mentioned some U.S examples such as Maryland and Chicago on Page 6 Line 251 to 252,

“Maryland took the lead by creating a statewide infiltration program during the mid 1980s, which became the origin of LID.”

and on Page 7 Line 291 to 293,

“The popular Green Alley programs in several U.S cities (e.g., Chicago and Los Angeles), although initially stormwater-focused, were acclaimed to provide ecological and cultural services.”

We would love to expand the work to examine the transition to more sustainable stormwater management at the city level. This paper, however, is mainly focused at the national and international scale and therefore would be complicated by introducing specific cities into the work. The paper is already a long read and for readability we suggest not adding many additional words.  

Point 4: Finally, are the USA and China the most progressive countries in sustainable Stormwater Management? Are there other cases worth to be mentioned and to some extent discussed? A discussion on why to compare the USA and China is already provided in the manuscript but what is lacking is a critical discussion on how these two countries fit in the global spectrum.

Response 4: Thank you for the interesting set of questions. Apart from the reasons that we have clarified in the text on Page 2 Line 112-117, we have chosen to compare China and the USA because they are typical representatives of two distinct social systems (capitalist and socialist) and they are arguably leading the way in terms of stormwater management. Of course, there are other countries that are also progressive such as Australia and our home New Zealand. We would love to include them into the conversation but feel this would add too much text to the paper. 

Point 5: Page 11 - Lines450: Not clear what Beijing 712 storm is.

Response 5:  Thank you for pointing out that this was not clear. We have altered the text on Page 11 Line 450 to clarify what is meant by 712 storm. The text now reads:

Although the Beijing storm disaster on July 12th was clearly an important SSWM transition node, it is important to appreciate the other contextual factors…
